# Distillation vs. Sampling for Efficient Training of Learning to Rank Models

Submission Id: 39

## ABSTRACT

In real-world search settings, learning to rank (LtR) models are trained and tuned *repeatedly* using large amounts of data, thus spending significant time and computing resources, and raising efficiency and sustainability concerns. One way to address these concerns is to reduce the size of training datasets. Dataset sampling and distillation are two classes of methods introduced to enable significant reduction in dataset size, while achieving comparable performance to training with complete data.

In this work, we perform a comparative analysis of dataset distillation and sampling methods in the context of LtR. We evaluate *gradient matching* and *distribution matching* dataset distillation approaches - shown to be effective in computer vision - and show how these algorithms can be adjusted for the LtR task. Our empirical analysis, using three LtR datasets, indicates that, in contrast to previous studies in computer vision, the selected distillation methods do not outperform random sampling. Our code and experimental settings are released alongside the paper.

## CCS CONCEPTS

• **Information systems** → Search engine architectures and scalability; *Retrieval efficiency*; *Learning to rank*.

## KEYWORDS

Dataset distillation, Learning-to-rank, Sampling

**ACM Reference Format:**
Anonymous Author(s). 2024. Distillation vs. Sampling for Efficient Training of Learning to Rank Models. In *Proceedings of the 2024 International ACM SIGIR International Conference on the Theory of Information Retrieval (ICTIR '24), July 23, 2024, Washington D.C., USA.* ACM, New York, NY, USA, 10 pages.

## 1 INTRODUCTION

Training a ranker at large scale is challenging due to the high complexity of the algorithms and the scale of real-world data. One way of addressing this challenge is by reducing the complexity of models and algorithms through quantization [40] or distillation [25, 35]. The alternative is to reduce the dataset size, either through dataset sampling [23] or dataset distillation [38]. The fundamental difference between these two techniques is that sampling *selects* the most valuable informative data points from the dataset, while distillation synthesizes a significantly smaller dataset with "combining" items from the original dataset.

Working with a reduced, possibly distilled dataset is beneficial for the following reasons: (i) Speed in training: a small, yet representative dataset allows for rapid development and investigation of new ideas. (ii) Energy efficiency in training: the reduced dataset will reduce the computational load, and decrease the energy consumption. Therefore, distillation is a greener solution and complies with green information retrieval (IR) initiatives [31]. (iii) Privacy: the full dataset may contain sensitive information or knowledge of a specific user's search behavior, possibly violating privacy guidelines. Moreover, a synthesized (i.e., distilled) dataset still contains essential information from the full data, without a one-to-one mapping to an actual user, thus contributing to privacy preservation.

Dataset distillation and sampling from datasets both lead to reducing the size of datasets in terms of number of data points, in contrast to dimensionality reduction approaches such as PCA which reduce number of features. However, they differ in how they generate smaller datasets. Sampling approaches *select* relevant data points based on a scoring function. In dataset distillation, new data points are *synthesized* and relevant information in the full data are *distilled* into these synthesized points.

Since the introduction of dataset distillation algorithms by Wang et al. [38], several variants have been proposed to enhance the effectiveness of distillation algorithms and optimize their complexity [30]. A key finding of prior work is that applying *dataset distillation allows for achieving considerably smaller datasets, while preserving the essence of the data*: models trained on the distilled data show comparable performance to models trained on the full dataset, and outperform models trained with datasets compiled via random sampling or core-set selection approaches [30, 32, 39]. The initial success stories of dataset distillation come from image classification [4, 9, 37, 43]. Further, Jin et al. [19] propose DasCond, which allows for distillation of graph-based datasets, and show that distillation is superior to sampling baselines on molecular datasets [17]. Similar findings have been observed with recommender systems. Recently, Sachdeva et al. [28] introduced a new recommendation model and applied dataset distillation to achieve a considerably smaller user-item interaction matrix.

Another key finding is that *the initialization of the distilled dataset has high impact on the convergence speed and ultimate performance achieved by the resulting distilled dataset* [8]. Last, but not least, by increasing the distilled dataset size, the performance of models trained on them increases [8, 30].

Inspired by these findings, we adapt dataset distillation for the learning to rank (LtR) task, and, in particular for feature-based LtR datasets. In this work, we present a comparative analysis between distillation and sampling for such datasets. Specifically, we investigate (i) whether ranking models trained with significantly smaller synthesized datasets through distillation algorithms can obtain comparable performance to the ranking performance of models trained with the full data, and (ii) whether dataset distillation algorithms

outperform random sampling baselines. To the best of our knowledge, we are the first to adapt gradient matching (GM) [43] and distribution matching (DM) [42] dataset distillation algorithms for the task of LtR by integrating LtR dataset with three simple and effective schemes.

In order to apply distillation methods, in our case GM and DM, it is not possible to directly plug a synthesized LtR dataset into existing implementations, which were originally designed for images [42, 43]. This is due to the complex and different structure of LtR datasets compared to images. To account for these differences, we discuss three strategies to construct a distilled dataset from a LtR dataset and elaborate how we apply distillation to LtR datasets via the GM and DM algorithms. These strategies allow for reducing the full data size, in a (i) query-wise, (ii) document-wise (or doc-wise), or (iii) label-wise manner, and construct an initial distilled dataset. The first two strategies are employed in GM whereas the latter is compatible with the DM algorithm.

The contributions of our study are as follows:

(C1) We propose three strategies to construct a distilled dataset for LtR data, and implement the GM [43] and DM [42] algorithms for LtR distillation.

(C2) By adapting distillation methods, we conduct experiments on three standard LtR datasets, namely MSLR-WEB30k [24], Istella-S [21], and Yahoo! Webscope [5] to investigate the following: (i) The applicability of dataset distillation to LtR datasets; (ii) The generalization of the following lessons from the dataset distillation literature for LtR datasets: (a) Dataset distillation methods outperform random sampling baselines. (b) Increasing the size of distilled dataset results in better performance at very high reduction ratios. (c) Initialization of distilled datasets affects the convergence speed and the quality of distilled dataset.

(C3) We provide a publicly available implementation of LtR distillation and the complete configuration of our experiments upon acceptance of this paper. [1]

The remainder of this paper is structured as follows. Section 2 introduces the LtR task and provides an overview of sampling and dataset distillation approaches. Then, we explain how we construct a distilled dataset and apply the GM and DM algorithms in Sections 3 and 4. Next, we describe our experimental setup and present our empirical analysis in Section 5. We conclude by summarizing our findings and discussing possible future directions in Section 6.

## 2 BACKGROUND AND RELATED WORK

### 2.1 Learning to rank

Suppose that $X_f$ is a feature-based LtR dataset, including multiple queries $\mathbf{Q} = \{q_i \mid i \in \{1, \ldots, N_Q\}\}$. For each $q_i$, there are $N_i$ candidate documents $D^{q_i} = \{v_{q_i d_j} \mid j \in \{1, \ldots, N_i\}\}$, where a query-document pair is represented by a feature vector $v_{qd}$ of length $S$ and a relevance score $r_{q_i d_j} \in 0, \ldots, R_{max}$ provided by experts. Given $\pi_\theta$ as the ranking model parameterized by $\theta$, $\pi_\theta(v_{q_i d_j})$ is the predicted relevance score of $q_i d_j$. The task of LtR is to optimize $\pi_\theta$ such that for any $d_p$ and $d_u$ in $D^{q_i}$ the following is achieved:

$$\forall q_i, r_{q_i d_p} > r_{q_i d_u} \Rightarrow \pi_\theta(v_{q_i d_p}) > \pi_\theta(v_{q_i d_u}). \tag{1}$$

---

[1] https://anonymous.4open.science/r/ictir-2024-7A2A/

### 2.2 Dataset sampling

Sampling from datasets has been widely studied in the literature to achieve smaller datasets for various goals. Initially introduced by Welling [39], herding is a method that allows for selecting nearest samples to a cluster center that has been used for incremental learning [26]. Another example is stratified sampling [6] for mining hard negatives in collaborative filtering with neural networks. Core-set selection methods based on [1] combined with active learning [27] are another line of research for finding representative subsets of datasets. In this direction, Sachdeva and McAuley [30], inspired by selection-via-proxy [7], propose a proxy model to score user interactions and sample from them to train collaborative filtering datasets. In addition to improving efficiency, dataset sampling has been used to improve the effectiveness of neural models by identifying better examples to use for training. For example, by clustering related queries together and then sampling queries from each cluster [16].

### 2.3 Dataset distillation

Wang et al. [38] introduced the task of dataset distillation in which they synthesize a new dataset and transform relevant information from a full dataset into it by their distillation algorithm. They achieve significantly smaller datasets with comparable performance on image classification tasks. Later, multiple variants of dataset distillation have been proposed. In this section, we take the generic formulation of dataset distillation suggested in a survey of distillation methods by Sachdeva and McAuley [30] to explain dataset distillation formally.

Let $X_f$ be the full dataset used to learn a task that contains $F$ data points represented by $S$ dimensional feature vectors. Then $X_d$ will be the distilled dataset that is represented through similar $S$ dimensional feature vectors with $D$ data points such that $D \ll F$. The task of dataset distillation is to transfer the useful relevant information from $X_f$ into $X_d$ such that applying a training algorithm $\beta$ on $X_d$ has comparable performance as on $X_f$ w.r.t. a predefined loss function $l$. Following [30], optimization in dataset distillation can be defined as:

$$\operatorname*{argmin}_{X_d} \left( \sup \left\{ \left\| l\left(\beta\left(X_f\right)\right) - l\left(\beta\left(X_d\right)\right) \right\| \right\} \right). \tag{2}$$

Moreover, Sachdeva and McAuley [30] classify the distillation methods into four categories: (i) gradient matching [43], (ii) distribution matching [42], (iii) meta-model matching [38], and (iv) trajectory matching [4]. Based on these primary methods, several updates have been introduced to improve dataset distillation. For instance, Jin et al. [19] demonstrate that applying gradient matching with graph neural networks (GNNs) for only one iteration suffices with graphs-based datasets, making distillation training faster because model weights do not need to be updated. Dataset distillation has been proven useful in federated learning [41] and privacy-friendly data sharing [11]. More recently, dataset distillation has been employed in user-item based recommendation to generate a smaller user-item matrix, and the recommendation performance of generated data is very close or in some cases better than on the full data [28]. Gu et al. [13] employ dataset distillation for continual learning and Such et al. [34] focus on accelerating neural architecture search by generating small synthetic data. Dataset distillation

has also been investigated for discrete data such as text. As an example, Maekawa et al. [22] apply distillation to fine-tune BERT [10] and Sachdeva et al. [29] introduce Farzi Data to summarize sequential data, such as text, into a smaller number of synthetic sequences.

**Distillation vs. sampling.** Several dataset distillation methods, including those highlighted in Section 2.1, are shown to outperform random sampling and core-set selection baselines [8, 30]. In particular, Cui et al. [8] show that (1) distillation performs best when the dataset size is reduced considerably, and (2) at lower reduction ratios, distillation and sampling methods perform similarly. In our work, we primarily focus on investigating the applicability of dataset distillation for the LtR task, and compare their effectiveness against sampling approaches. Additionally, we examine whether increasing the size of distilled datasets is beneficial for performance and if initialization of the distilled dataset influences the effectiveness and convergence speed of distillation as suggested by previous work [8, 19, 30, 42, 43]. To this end, we select two representative variants of distillation algorithms that are effective and scalable. We choose a distribution matching (DM) by Zhao and Bilen [42] and a gradient matching (GM) algorithm by Zhao et al. [43]. In particular, we choose a GM algorithm as it makes the overall optimization distillation compared to meta-model matching methods more efficient [30], and trajectory matching approaches are costly since they require fully computing the training trajectories of full and distilled datasets. To the best of our knowledge, we are the first to adapt GM and DM dataset distillation for LtR datasets.

## 3 STEP 1: CONSTRUCTING AN INITIAL DISTILLED DATASET

Let $X_f$ be a LtR dataset. We examine two variants of dataset distillation methods, gradient matching [43] and distribution matching [42], to perform the optimization in Eq. (2), where the learning algorithm in (2) addresses the LtR task explained in Eq. (1). To applying a distillation algorithm, two stages should be considered:

(1) Construct an initial distilled dataset $X_d$ based on $X_f$.
(2) Apply a distillation algorithm and calculate the distillation loss $l_{dist}$ and update $X_d$ to minimize the distillation loss.

To create an initial distilled dataset $X_d$, we propose query-wise, doc-wise, and label-wise initial construction of the distilled dataset in this section. Then, in Section 4, we explain the second step; how dataset distillation for LtR based on gradient matching (Section 4.1) and distribution matching (Section 4.2) is performed.

We apply distillation to a synthesized dataset $X_d$ that is constructed prior to algorithm execution. $X_d$ resembles the full dataset $X_f$ in feature dimension but has considerably fewer data points, in our case feature vectors of query-document pairs. In the case of distillation for fully supervised tasks such as image classification, the initial distilled dataset $X_d$ is simply constructed by synthesizing randomly initialized images per each class. This strategy is not suitable for an LtR dataset that is characterized by several queries and their candidate documents with relevance scores:

(1) Although queries and documents across the dataset are different from each other, in practice queries could be very similar so that one document might be relevant to various queries at different levels. Accordingly, some feature vectors across the dataset might be very similar to each other yet have different relevance scores.
(2) Even though some LtR datasets contain relevance scores for each query-document pair, the distribution of relevant documents for two distinct queries may be different so that grouping of query-document pairs merely based on relevance labels is not reasonable.

The optimal structure for constructing an initial distilled dataset $X_d$ for LtR requires allocating a number of queries, a number of their associated documents, and specifying relevance scores for query document pairs given a certain dataset size. In the absence of prior work in this space, we make some very natural first choices: the chief elements of an LtR dataset are the queries, documents, and relevance scores. Based on this, we propose three strategies, namely, query-wise, document-wise, and label-wise construction of a distilled dataset $X_d$. The first two variants are employed in the gradient matching distillation algorithm and the last one is suitable for distribution matching distillation, as we will see below. After constructing the distilled datasets through these strategies, all features are randomly initialized.

The proposed strategies to create an initial distilled dataset $X_d$ can be implemented in multiple ways. Yet implementing them such that considering a given size of distilled data, all yield distilled datasets of the same size is challenging. We observed that the ranking performance does not linearly vary based on the exact number of query-document pairs. Hence, we construct the distilled datasets $X_d$ subject to a *distillation budget* $\mathcal{Z}$, which is a rough estimate of the distilled dataset's size: constructing distilled datasets with the same $\mathcal{Z}$ for each of the proposed strategies leads to different dataset sizes. Higher values of $\mathcal{Z}$ indicate smaller reduction in dataset size for all strategies. Moreover, queries with the query-wise strategy, documents within each query with the doc-wise strategy, and documents within each query and relevance label with the label-wise strategy are selected randomly, and hence, these strategies are not deterministic. Figure 1 illustrates an example LtR dataset and how the initial distilled dataset is constructed by the proposed strategies. The example dataset includes 5 queries and binary relevance scores where grey-colored vectors are non-relevant and green-colored vectors are relevant. The distilled datasets are constructed by applying query-wise, doc-wise, and label-wise strategies with distillation budgets $\mathcal{Z}$ of 0.4, 0.4, and 1, respectively.

*Query-wise strategy.* In this strategy, we reduce the size by keeping only a selection of queries and discarding the rest while preserving all of the documents for selected queries and their relevance scores. $\mathcal{Z}$ is the approximate ratio of $\frac{|X_d|}{|X_f|}$. The selected queries in $X_d$ are chosen randomly one-by-one until $|X_d| > \mathcal{Z} \cdot |X_f|$ is satisfied. As an example, in Figure 1 the size of $X_f$ is 33 and consider $\mathcal{Z} = 0.4$. Accordingly, $\lfloor \mathcal{Z} \cdot |X_f| \rfloor = 13$ and the construction of $X_d$ after randomly selecting $q_1$ and $q_3$ is finished since $|X_d| = 15$.

*Doc-wise strategy.* The doc-wise strategy tries to keep all the queries but reduces the number of available documents per query. To this end, $\mathcal{Z}$ is the percentage of documents randomly selected per query regardless of their relevance scores. We ignore relevance scores in this strategy, since in the standard LtR datasets, the number of relevant documents per query is limited and per query and per relevance selection of documents causes the complete removal of

Full LtR dataset

Figure 1: Constructing a distilled dataset from an example LtR dataset (top). From left to right, we show distilled datasets that have been constructed by employing query-wise, doc-wise, and label-wise strategies with a $0.4$, $0.4$, and $1$ distillation budgets, respectively.

relevant documents. Conversely, since the majority of documents per query are irrelevant, most of the discarded documents will be irrelevant ones and highly relevant documents are more likely to be preserved. Furthermore, some queries have only few available documents. As a consequence of a large reduction factor per query, all of a query's documents may be removed; those queries will not be present in the constructed distilled dataset. As further illustration, with the example LtR dataset in Figure 1, employing a doc-wise strategy with $\mathcal{Z} = 0.4$ results in a distilled dataset with all the queries from the full dataset but fewer documents per query.

*Label-wise strategy.* This strategy is similar to the doc-wise strategy as all queries are preserved and the number of documents per query are reduced considerably but differs from doc-wise in considering relevance scores. With the label-wise strategy, we consider a minimum of $\mathcal{Z}$ document(s) for each existing relevance label in each query. Thus, if a query has $Z_r$ documents with relevance score of $r$, there would be $\min(Z_r, \mathcal{Z})$ documents considered in the initial considered. In the simplest way, shown in Figure 1, $\mathcal{Z} = 1$ and since $q_1$ to $q_5$ all have both relevant and non-relevant documents, one of each exists for all queries in the constructed distilled dataset.

## 4 STEP 2: DISTILLATION ALGORITHMS FOR LEARNING TO RANK

Having defined multiple ways of obtaining an initial distilled dataset $X_d$, we adapt and apply distillation algorithms to it.

### 4.1 Gradient matching distillation

To start, we adopt the gradient matching distillation approach introduced by Zhao et al. [43]. In this approach, a deep neural network parameterized by $\theta$ is trained with both the distilled and full datasets; then, the training loss for both datasets is calculated separately $(l_d, l_f)$. After which, the gradients of the model are derived w.r.t. to these losses $\nabla_\theta l_d, \nabla_\theta l_f$ and $X_d$ is optimized such that the gradients from distilled data $\nabla_\theta l_d$ matches gradients from original data $\nabla_\theta l_f$ over multiple iterations. Intuitively, if $X_d$ is representative of $X_f$, then models trained on it should reach the same optimization point

---

**Algorithm 1** Gradient matching distillation

1: Construct initial $X_d$ by query-wise or doc-wise strategies.
   ▷ As explained in Section 3.
2: **for** $k = 0, 1, \ldots, K - 1$ **do**
3:     Sample $\theta_k \sim P_\theta$
4:     **for** $i = 0, 1, \ldots, E - 1$ **do**
5:         **if** *query_wise* **then**
6:             $X_d = \text{GRADIENTMATCHING}(\theta_k, X_d, X_f^b), \ \forall X_f^b \in X_f$
                     ▷ Multi-batch distillation of $X_f$.
7:         **else if** *doc_wise* **then**
8:             $X_d^b = \text{GRADIENTMATCHING}(\theta_k, X_d^b, X_f^b),$
                   $\forall \left(X_d^b, X_f^b\right) \in \left(X_d, X_f\right)$
         ▷ One-to-one multi-batch Distillation of $X_d$ and $X_f$.
9:         **end if**
10:         $opt\left(\theta_k \mid \nabla_{\theta_k} l_r\left(\pi_{\theta_k}\left(X_d\right), r_{X_d}\right)\right)$
                   ▷ Update model weights $\theta_k$.
11:     **end for**
12: **end for**
13: **procedure** GRADIENTMATCHING$(\theta_k, X_d^b, X_f^b)$
14:     $l_f^b = l_r\left(\pi_{\theta_k}\left(X_f^b\right), r_{X_f^b}\right)$   ▷ Ranking loss with original data
15:     $l_d^b = l_r\left(\pi_{\theta_k}\left(X_d^b\right), r_{X_d^b}\right)$   ▷ Ranking loss with distilled data
16:     $l_{dist} = l_{gmd}\left(\nabla_{\theta_k} l_f^b, \nabla_{\theta_k} l_d^b\right)$
                 ▷ Gradient matching distillation loss.
17:     $opt\left(X_d^b \mid \nabla_{X_d^b} l_{gmd}\right)$          ▷ Update $X_d^b$
18:     **return** $X_d^b$
19: **end procedure**

---

as the models trained on the full data.

Algorithm 1 depicts the proposed gradient matching distillation adjusted for LtR datasets. Initially, as explained in Section 3, we construct a distilled dataset based on the query-wise and doc-wise strategies (line 1). In our case, $\pi_\theta$ will be the deep neural network

that functions as LtR model. Similar to [43], we initialize model weights $K$ times and employ them in distillation to ensure that $X_d$ generalizes for various model initializations (line 3). Next, if $X_d$ is constructed by the query-wise strategy, we distill batches of the original dataset one by one into the single batch of distilled data (lines 5, 6). In the case of the doc-wise strategy, $X_d$ and $X_f$, both include several queries though $X_d$ having considerably fewer documents, hence, each batch of original data is distilled into a corresponding batch of distilled data (line 7, 8). Although according to Jin et al. [19], it is possible to perform gradient matching for each set of initialized weights $\theta_k$ only once, we follow the original variant of gradient matching [43] by updating model weights $\theta_k$ (line 10) after each iteration of gradient matching distillation (lines 6, 8) for multiple epochs $E$.

We follow the gradient matching procedure from [43] but adjust for usage in the LtR domain. In the original gradient matching, images are distilled in the groups with the same label. In the case of LtR, query-document pairs from different queries with the same relevance labels do not correlate with each other and cannot be seen as similar entities. We preserve the conventional LtR training pipeline and feed the original and distilled data $X_f^b, X_d^b$ in batches of multiple queries into $\pi_{\theta_k}$ and calculate corresponding losses $l_f^b, l_d^b$ through a ranking loss $l_r$ (lines 14, 15). Then, the distillation loss $l_{dist}$ calculates the distance between $\nabla_{\theta_k} l_f^b$ and $\nabla_{\theta_k} l_d^b$ with $l_{gmd}$ (line 16) and is employed to optimize and update $X_d^b$ (line 17). We adopt the mean squared error (MSE) loss to calculate $l_{gmd}$ between all corresponding pairs of $\nabla_{\theta_k} l_f^b$ and $\nabla_{\theta_k} l_d^b$.

## 4.2 Distribution matching distillation

Next, as an alternative, we build on the method introduced by [42] for dataset distillation based on distribution matching that is illustrated in Algorithm 2. In this approach, distilled and full data feature representations are first transformed into a latent space through $K$ randomly initialized encoders $\phi_k$. Then, the distillation loss $l_{dist}$ is calculated based on a maximum mean discrepancy (MMD) criterion [12], which has been used to calculate the distance between two data distributions in core-set selection methods. In this approach, the distribution of distilled data is optimized to be similar to the original data.

In the original approach [42], MMD is calculated within classes, i.e., distributions of the distilled dataset $X_d$ and full dataset $X_f$ are compared for each class. As explained in Section 4.1, relevance labels in LtR cannot be associated in the same manner as class labels of image datasets are in dataset distillation, but we can assume the distribution of feature vectors per query and for each relevance score to be similar. Therefore, to employ distribution matching for LtR datasets, we only use the label-wise strategy since it facilitates one-to-one distribution matching of feature vectors per query and per relevance label between the full and distilled dataset. Accordingly, after initial construction, $X_d$ and $X_f$ contain the same number of queries and the same number of unique relevance labels per query.

In the distribution matching procedure, batches of $X_d^b$ and $X_f^b$ are encoded by $\phi_{\theta_k}$ into latent representations $\phi_f^b, \phi_d^b$ (lines 7, 8).

---

**Algorithm 2** Distribution matching distillation

1: Construct initial $X_d$ by label-wise strategy.
  ▷ As explained in Section 3.
2: **for** $k = 0, 1, \ldots, K - 1$ **do**
3:   Sample $\theta_k \sim P_\theta$
4:   $X_d^b = \text{DistributionMatching}(\theta_k, X_d^b, X_f^b),$
      $\forall \left( X_d^b, X_f^b \right) \in \left( X_d, X_f \right)$
      ▷ One-to-one multi-batch Distillation of $X_d$ and $X_f$.
5: **end for**
6: **procedure** $\text{DistributionMatching}(\theta_k, X_d^b, X_f^b)$
7:   $\phi_f^b = \phi_{\theta_k} \left( X_f^b \right)$        ▷ Encoding original data
8:   $\phi_d^b = \phi_{\theta_k} \left( X_d^b \right)$        ▷ Encoding distilled data
9:   $l_{dist} = l_{dmd} \left( \phi_f^b, \phi_d^b \right)$
      ▷ Distribution matching distillation loss
10:   $opt \left( X_d^b \mid \nabla_{X_d^b} l_{dist} \right)$        ▷ Update $X_d$
11: **end procedure**

---

Then $l_{dist}$ is derived with a distribution matching loss $l_{dmd}$, which is based on MMD (line 9). For $l_{dmd}$ we follow the definition in [42], as given in Eq. (3). Finally, we update $X_d^b$ by calculating its gradient w.r.t. $l_{dist}$ (line 10):

$$l_{dmd} = \left\| \frac{1}{| X_f^b |} \sum_{X_f^b} \phi_f^b - \frac{1}{| X_d^b |} \sum_{X_d^b} \phi_d^b \right\|. \quad (3)$$

## 5 EVALUATION

In our empirical analysis, first, we validate the applicability of dataset distillation, i.e., the modified GM and DM algorithms, for LtR datasets. Then, we examine the performance of distillation methods for LtR datasets w.r.t. to the following aspects:

(A1) The influence of distilled datasets initialization on the convergence speed and quality of distilled datasets.
(A2) The effect on increasing the size of distilled datasets at very high reduction ratios on ranking performance.
(A3) The performance of distillation methods in comparison to random sampling baselines.
(A4) The impact of ranking loss in the context of datasets distillation.

## 5.1 Experimental setup

**Datasets.** We perform our experiments on the following publicly available datasets: MSLR-WEB30k [24], Istella-S [21], and Yahoo! Webscope [5]. MSLR30K contains 31, 531 queries, each including multiple documents with relevance labels from 0 to 4. We use the standard train/validation/test set of the first fold in MSLR30K dataset that contain 60%, 20%, and 20% of queries, respectively. Each query-document pair is represented by a 136 dimensional feature vector and a relevance label. Istella-S includes slightly more queries than MSLR30K, 33, 018 queries with 220 features representing each query-document pairs. The standard split sets in this dataset are created with similar ratios as MSLR30K. Finally, the Yahoo! Webscope dataset contains 29, 921 queries, and the standard training sets are

**Table 1: Statistics of the datasets after preprocessing.**

| Dataset | Query-document vectors | | | Avg # | Avg. % |
|---|---|---|---|---|---|
| | Train | Valid. | Test | docs/query | $r > 0$ docs |
| MSLR30K | 2,258,066 | 743,354 | 749,217 | 123.3 | 46.1 |
| Istella-S | 2,042,342 | 684,076 | 679,749 | 106.2 | 13.1 |
| Yahoo! | 466,687 | 70,054 | 163,479 | 24.3 | 77.9 |

generated by splitting the dataset with 67%, 10%, and 23% ratios and query-document pairs are represented with 699 dimensional feature vectors. Table 1 shows the total number of query-document vectors after pre-processing these datasets in train, validation and test sets. The last two columns include the average number of documents per query and the average percentage of documents with non-zero relevance labels per query in order.

We used all three datasets in our experiments as they differ in distribution of relevance labels, number of features, and size to cover a variety of data settings.

**Implementation details.** We use Pytorch-ltr [18] for training LtR models and preprocessing of the datasets. The preprocessing of datasets, consists of filtering queries without any relevant documents for all datasets and as Yahoo! is already normalized, we only apply per query normalization for MSLR30K and Istella-S datasets.

In our implementations, we follow the GM [43] and DM [42] implementations while adjusting for LtR datasets as explained in Section 3. We use MSE as a point-wise ranking loss and gradient distillation loss (line 16 in Algorithm 1), and MMD for distribution matching loss (line 9 in Algorithm 2) [42, 43]. Moreover, we study the effect employing other ranking loss functions such as LambdaLoss and observe similar conclusions in our analysis. Finally, since during dataset distillation, the gradients of the LtR requires back-propagation, we do cannot employ tree-based implementations such as LambdaMART [20]; therefore, similar to [2, 14, 36], we use a three layer multi-layer perceptron (MLP) as the ranking model $\pi$ in Algorithm 1 and a two layer MLP as the encoder $\phi$ in Algorithm 2.

The training loop in Algorithm 1 consists of $E = 10$ (line 4 in Algorithm 1) epochs and optimization of $\theta_k$ (line 10 in Algorithm 1) consists of 20 epochs. We use a batch size of 256 and set the learning rate to 0.01 for updating the distilled dataset and to 0.1 for training ranking models in all experiments. Moreover, we set the parameter $K$ to the number of different model initializations, i.e., the number of distillation iterations, (line 2 in Algorithm 1, line 2 in Algorithm 2) at 500. We found that employing early-stopping with validation sets while training the distilled dataset is highly beneficial; thus, while training we early-stop the training both within each iteration and across all iterations if no improvement based on validation performance is achieved after certain number of steps.

We execute each setting 5 times, i.e., we create 5 distilled datasets in each setting. We use a two-tailed student's t-test for significance testing [33] and apply Bonferroni correction [3] using a significance level of $\alpha = 0.0001$. In result figures the shadow area represents 90% confidence intervals. To evaluate performance of the LtR model, we measure NDCG@10 and average relevant position (ARP) in all experiments. Since the same conclusions were derived from ARP results, we only include them in the released repository of the paper.

**Evaluation pipeline and baselines.** In total, we have 3 (algo-

rithms) * 2 (initializations) = 6 ways of constructing and training a distilled dataset. We implement gradient matching with the query-wise strategy GM-Q and the doc-wise strategy GM-D, and distribution matching with the label-wise strategy DM-L. In terms of distilled dataset initialization, we consider random initialization of the feature vectors in the initial constructed distilled dataset -R or use real feature vectors from the full LtR dataset -NR. Lastly, only for in the context of comparing loss functions, additional -P, and -L suffixes imply the use of MSE and LambdaLoss, respectively.

The distillation experiments pipeline is organized as follows. First, we construct the initial distilled dataset from the training set of the full dataset. Then, based on the distillation algorithm, we train the initial distilled data using the full dataset. Finally, we use the trained distilled dataset to train LtR models and evaluate their performance using the test set from the full dataset. We compare these methods to the following baselines: (i) query-wise random sampling (QS), (ii) document-wise random sampling (DS), (iii) label-wise random sampling (LS), and (iv) full dataset (Full). For the sampling baselines, we sample from the full dataset based on those strategies and then train LtR models with the sampled dataset to evaluate their performance with the full dataset test set. Full dataset performance is the maximum possible performance that can be achieved.

We display the outcomes of our experiments in Figures 2–5. We show the performance of distillation algorithms and sampling baselines for various dataset sizes using NDCG@10. Section 3, we defined $\mathcal{Z}$ to explain how the distilled datasets are initially constructed. In the GM experiments, we consider $\mathcal{Z} = 0.005$–$0.05$ and for the DM experiments $\mathcal{Z} = 1$–$6$. For improved clarity, the x-axis of the figures shows the relative size of distilled or sampled datasets in comparison to the full dataset size for the corresponding $\mathcal{Z}$ value. The points on the x-axis for different datasets differ within the same strategy since the datasets have different numbers of queries, documents and distribution of relevance scores. Since having very small values of $\mathcal{Z}$ with the doc-wise strategy on the Yahoo! and Istella-S datasets results in the removal of all documents from the full datasets, the construction of an initial distilled dataset is not feasible. That results in fewer points on the x-axis for those experiments.

## 5.2 Empirical analysis

**Applicability of dataset distillation for LtR datasets.** To investigate if dataset distillation, in particular the GM and DM algorithms, can be applied for LtR datasets, we first train LtR models with the distilled dataset right after their initial construction (Section 3) and measure the model performance with the full dataset test set. Then, we apply the distillation algorithm on the distilled datasets, and after training new LtR models with them, we measure their test performance. We compare these test performances for various $\mathcal{Z}$ values across all datasets and illustrate them in Figures 2–4. Particularly, we are interested in comparing the gold (the performance after applying distillation) and red lines (the performance before applying distillation, i.e., the performance of random vectors (RV)).

The results indicate that with the GM algorithm, both query-wise and doc-wise strategies are successful in distilling useful information from the full dataset into the distilled dataset across all $\mathcal{Z}$

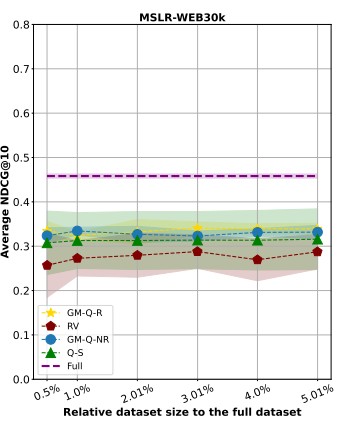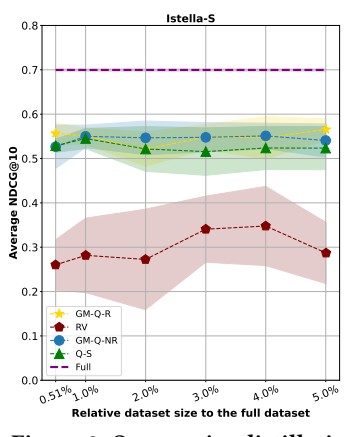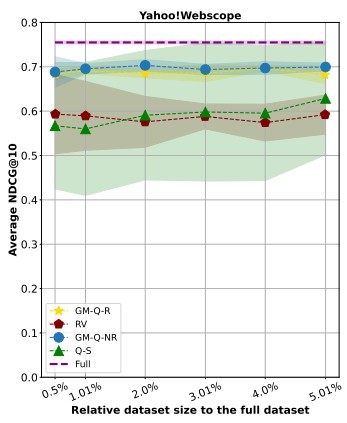

**Figure 2: Query-wise distillation.**

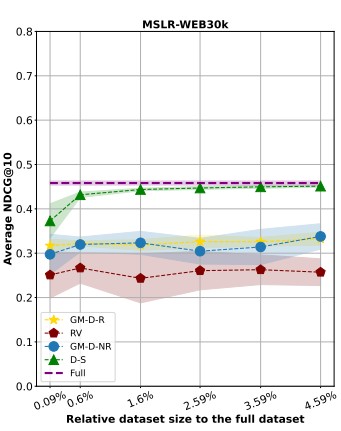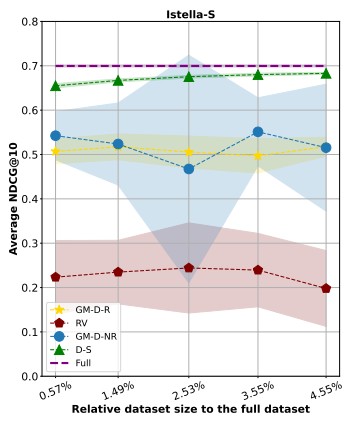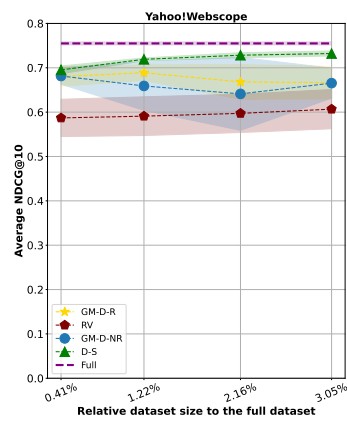

**Figure 3: Doc-wise distillation.**

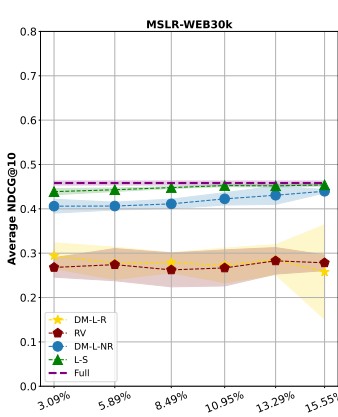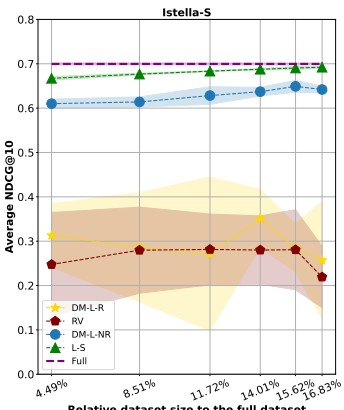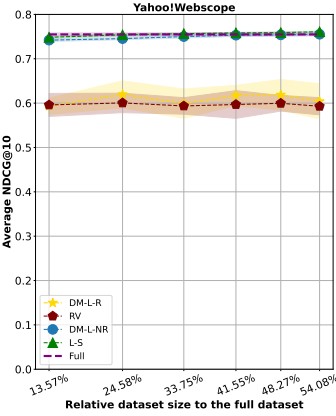

**Figure 4: Label-wise distillation**

values and all datasets. On the contrary, applying the DM algorithm with label-wise strategy is not successful and even though there are a few cases such as Istella-S at 14.01% ratio where training with DM results in some minor improvements, this difference is insignificant. As expected, RV performance has higher variability than distilled dataset performance, and the variance of GM results is overall smaller for the MSLR30K dataset. We conclude that dataset distillation can be employed for LtR datasets given the appropriate initial construction of the distilled dataset.

**(A1) Effect of initialization on quality and convergence speed.**

**Table 2: Average number of iterations spent to complete distillation training.**

| Initialization | Query-wise | Doc-wise | Label-wise |
|---|---|---|---|
| Random | 20.2 | 28.3 | 18.4 |
| Real | 18.3 | 11.1 | 17.7 |

In their benchmark of distillation methods, Cui et al. [8] demonstrate that the initialization of distilled datasets affects the convergence speed of distillation training and the final performance on the distilled dataset. They conclude that using K-Center [15] to initialize the distilled dataset instead of random initialization results in faster convergence and increases performance in some settings. In our work, we compare two categories of initialization: *random initialization* and *randomly sampled datasets* based on query-wise, doc-wise, and label-wise strategies. To analyze the performance results, we compare the blue curves (note the -NR suffix) with the gold curves (random initialization) and the red ones (RV) in Figures 2–4.

The results indicate that initializing from actual vectors of the dataset (randomly sampled datasets) substantially benefits the label-wise DM algorithm across all datasets, where distillation performance is significantly better than RV and randomly initialized distilled datasets. However, we do not observe any meaningful differences with the query-wise and doc-wise GM settings, though there are minor boosts at some of the $\mathcal{Z}$ values.

In Table 2, we show the number of iterations it took for the distillation algorithm to early-stop. Overall, we observe that using real data points instead of random initialization is faster especially for doc-wise GM. Our observations w.r.t. convergence speed aligns with the previous work [8]. However, w.r.t. performance of the distilled dataset, improvements were only observed in label-wise settings with the DM algorithm.

**(A2) Effect of increasing the size of distilled datasets.** Larger distilled datasets lead to improved performance [30], an expected behaviour given that more data usually improves accuracy. Cui et al. [8] investigate this and conclude that distillation performance increases with very high reduction ratios, but it behaves similarly to random sampling when the reduction of the full dataset is not large. We investigate this finding for very high reduction ratios by training distilled datasets of various sizes at very high reduction ratios, i.e., small values of $\mathcal{Z}$ and illustrate them in Figures 2–4.

As opposed to previous work with different datasets, we do not observe the same trend for LtR datasets in any of our distillation methods. We also notice that the performance of sampling approaches has only mild improvements and is very close to the full dataset performance.

**(A3) Distillation vs. sampling.** Several works have shown that distillation outperforms random sampling baselines with different types of datasets [8, 30, 42, 43]. We investigate whether this finding generalizes to LtR datasets. To this end, we consider the performance of sampling baselines, namely QS, DS, and LS (the green curves in Figures 2–4) and compare them with corresponding distillation approaches (gold and blue curves), as each strategy leads to distinct dataset sizes. We also provide an overall comparison of distillation (in the setting where the distilled dataset is randomly initialized) and sampling approaches in Table 3.

Consider Figures 2–4 again. Starting from query-wise GM, we notice that distillation outperforms QS on average across all datasets, but the results of these methods are within their confidence intervals so the advantage is not significant. Additionally, we observe that QS results have high variance, particularly with the Yahoo! dataset where including some queries has had a large impact on the performance of LtR models. For the other two variants of distillation, we notice that both DS and LS methods outperform distillation methods consistently while having low variance for all datasets at various dataset sizes. The only exception is DM on the Yahoo! dataset, where LS and DM-L-NR have very similar performance. Also, we notice a high performance with Yahoo! even with the RV method. We suspect this is due to the fact that Yahoo! has few documents per query and the number of documents with non-zero relevance scores (see Table 1) is much higher than other datasets.

To illustrate the results in Table 3, we select the best performing datasets obtained by sampling and distillation using validation sets, and compare their full dataset test performance. In the label-wise DM settings, even for small values of $\mathcal{Z}$, the generated datasets are larger than the generated datasets with the other two strategies. Since we use the same test in all experiments and increasing the dataset size does not affect the performance considerably, we do not consider the size of datasets in overall comparison of methods in Table 3. In terms of distillation methods, GM distillation performs better than DM distillation for all three datasets, while the query-wise strategy and the doc-wise strategy perform similar to each other, with a slight difference in favor of the query-wise strategy for the Istella-S datasets. The reason that doc-wise and label-wise strategies for Istella-S perform noticeably weaker compared to other datasets is probably due to the low percentage of documents with non-zero relevance labels (see Table 1). In terms of sampling approaches, LS performs better than DS, which performs better than QS. The sampling methods are generally achieving a very high ranking performance, given the small data sizes they have.

In conclusion, in contrast to previous findings [8, 30], we observe that neither the GM nor the DM algorithm outperforms random sampling baselines. They do not achieve comparable performance to the full dataset in most of the settings. We believe this is due to the fact that LtR datasets have a lot of redundancy in them, and in turn, even very aggressive sampling of datasets, does not hurt the full data performance considerably. Additionally, the similarity between feature vectors per query and across all queries makes distillation of valuable and useful information from the full dataset into the distilled dataset challenging.

**(A4) MSE vs. LambdaLoss.** In our analysis in A1 -A4, we employed MSE as the ranking loss. Here, we investigate if our observations, particularly that random sampling baselines outperform dataset distillation approaches, generalize with LambdaLoss as the ranking loss. We continue with GM-Q because it achieves better performance than the other two variants, and in the context of LambdaLoss, having enough documents with various relevance labels is needed. Note that some queries in the MSLR30K dataset contain up to 1251 documents; due to memory limitations in our infrastructure, we remove queries with more than 300 documents (around 2.3%), and reduce the batch size to 64.

In Figure 5, we compare MSE and LambdaLoss when training with full datasets (Full-P, Full-L), query-wise randomly sampled

**Table 3: Comparison of the best results from distillation and sampling, per dataset. with NDCG@10. Best results per class of methods (distillation or sampling) in bold, best overall result underlined. † denotes a significant difference between the best distillation method and the rest of results for each datasets. Similarly, ‡ indicates a significant difference between the best sampling method and the rest.**

| Dataset | GM-Q | GM-D | DM-L | QS | DS | LS | Full |
|---------|------|------|------|----|----|----|----|
| MSLR30K | **0.351**‡ | 0.350‡ | 0.350†‡ | 0.349‡ | 0.453† | **0.456**† | 0.463† |
| Istella-S | **0.569**‡ | 0.503†‡ | 0.181†‡ | 0.558‡ | 0.685† | **0.694**† | 0.703† |
| Yahoo | 0.692‡ | **0.694**‡ | 0.584†‡ | 0.713†‡ | 0.737†‡ | **0.761**† | 0.760† |

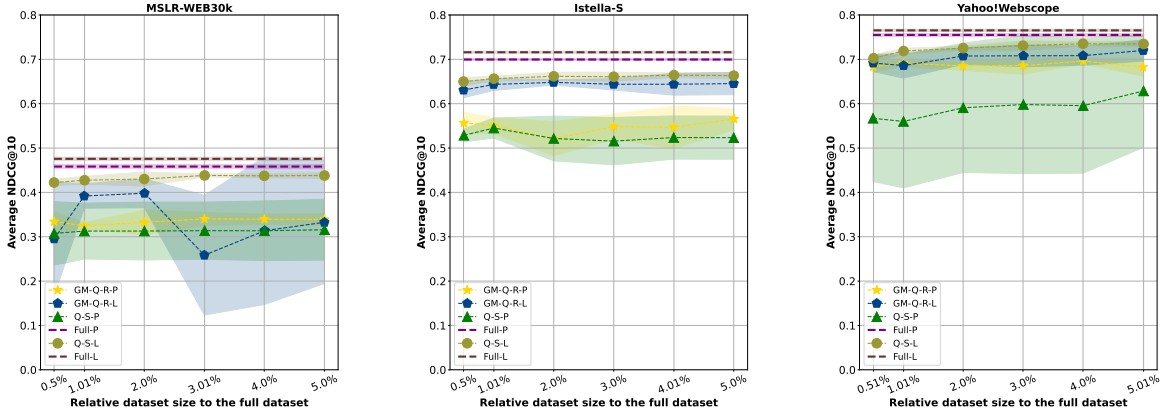

**Figure 5: Comparing the distillation performance between MSE and LambdaLoss.**

datasets (Q-S-P, Q-S-L), and query-wise distilled datasets (GM-Q-R-P, GM-Q-R-L). As expected, LambdaLoss leads to better ranking performance w.r.t. NDCG in all experiments, including improving the ranking performance for distilled datasets. However, the randomly sampled datasets outperform the distilled datasets. Employing MSE in query-wise distillation gained some minor improvements over query-wise random sampling (and not other random sampling baselines); conversely, query-wise random sampling with LambdaLoss outperforms GM-Q with LambdaLoss. This reinforces our primary conclusion that random sampling baselines outperform dataset distillation methods for the task of LtR.

## 6  CONCLUSION AND FUTURE WORK

In search for more efficient training of LtR models, we investigate the merits of distillation versus sampling as techniques for dataset reduction. Previous work has demonstrated the effectiveness of dataset distillation methods for various types of datasets, including images, graph-based data, and in user-item based recommendation datasets. In this work, we evaluated two primary distillation algorithms - distribution matching (DM) and gradient matching (GM) - for LtR datasets. To make existing distillation techniques compatible to LtR datasets, we introduced three strategies, namely query-wise, doc-wise, and label-wise, to construct an initial distilled dataset.

To put distillation in perspective, we compared its results against sampled datasets. We further evaluated both the distillation and sampling techniques, focusing on assessing the ability of the reduced dataset to train the model to a similar performance level as the original dataset.

Our findings indicate that, in contrast to prior work on distillation with different types of data [19, 28, 43], dataset distillation for LtR does not outperform simple sampling baselines. Also different from previous results, the distilled LtR datasets created at very high reduction rates do not improve the performance in most of the settings. Finally, we observe an important similarity to prior work on distillation: the initialization of distilled datasets affects the convergence speed of distillation algorithms for LtR, though initializing by real feature vectors only brings performance improvements in the label-wise DM setting.

Meanwhile, our sampling results suggest that a high degree of redundancy exists in LtR datasets; thus, the environmental and economic cost of research and development in LtR could be decreased by using smaller, more representative datasets. Additionally, revisiting and enriching features in LtR datasets would be highly beneficial to achieve further improvements in LtR models.

We note that our adaptation of GM and DM algorithms is limited in how an initial distilled dataset is constructed as our proposed strategies are not guaranteed to be the optimal strategy.

We plan to expand our analysis to very large, real-world datasets, to determine how dataset-specific our current results are. Moreover, adding new dataset distillation approaches, like meta-model matching and trajectory matching would be interesting.

As our empirical evaluation shows sampling performing very well, we expect the state-of-the-art datasets contain, in fact, significant redundancy. One direction of future work is to define (and validate) a metric for redundancy, analyze possible sources of redundancy in LtR datasets, and asses the feasibility of reducing it. Moreover, further investigations w.r.t. the relation between features and the performance of sampling and distillation approaches should bring additional insights into the quality of LtR datasets and how they could be improved in the future.

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
