# OpenReview forum: "Distillation vs. Sampling for Efficient  Training of Learning to Rank Models"
_ACM.org/SIGIR/ICTIR/2024/Conference — ICTIR 2024_

### Official Review · Reviewer_Dzin · 2024-05-14

**Rating:** 0
**Confidence:** 4

**Objective Part Of Review:**

(1) Is the problem clearly stated?

Yes, this paper empirically evaluates dataset distillation's effectiveness in learning to rank tasks, compared against the random sampling method.

(2) Are the methods clearly described?


For improved clarity, Section 3 and 4 should include diagrams to outline the model training process and clearly marke data representations before and after distillation.

(3) Are the results clearly stated?

Yes, the paper provides corresponding analysis results on three LTR datasets.

(4) Are the various claims in the paper supported?

Most claims are supported.

(5) Is each concept and notation properly defined before it is used?

Yes.

(6) Can the abstract and the introduction be understood before one has read and understood the rest of the paper?

Yes.

(7) Did you spot any contradictions or other signs that something is wrong?

No.

(8) Is work that is directly relevant or even competing with the work in this paper cited?

Yes.

(9)Provide falsifiable evidence for your criticism:

My main concern is whether the results, derived from a simple LTR model such as the three-layer MLPs, will hold consistently for more advanced or complex LTR models.

**Subjective Part Of Review:**

(1) Did you find the paper easy to read and understand?

Yes, the paper is easy to follow.

(2) Do you find the problem relevant?

Yes.

(3) Do you find the methods original?

This paper offers some original contributions to the field.

(4) Do you find the results interesting?

The results are interesting; however, there is a concern regarding their adaptability to other LTR models.

(5) Do you think that others in the ICTIR community will be interested in this work?

Have some interests.

---

### Official Review · Reviewer_xLYE · 2024-05-16

**Rating:** -1
**Confidence:** 4

**Objective Part Of Review:**

## Soundness:

The general concept of the paper—applying dataset distillation to Learning to Rank (LtR)—is intriguing and somewhat novel, as it has not been previously explored. However, the motivation behind dataset distillation, the methods used, and the claims made are not clearly articulated. For example:
- Why is dataset distillation effective for other tasks (such as image processing, graph analysis, and recommendation systems), and do these reasons apply to IR as well?
- Are we first identifying a subset (X_d) from the full dataset (X), and then training the LtR model solely on X_d? Or are these two steps performed simultaneously?
- In both scenarios, aspects such as "speed of training" and "energy efficiency of training" are not addressed at all, or possibly even worsened; additionally, the privacy aspect was not discussed at all. Therefore, the overall motivation for dataset distillation remains unclear to me.

## Presentation:

Partly due to the unclear problem definition and motivation, the methodologies described in this work are difficult to understand. The figures provided some clarification, but the algorithms did not. Sections 3 and 4 should be comprehensible independently, without the need to rely on figures, algorithms, or other data distillation papers. The current form of presentation falls short of this standard.

## Difficulty:

A decent amount of work has been conducted on an interesting problem. No complaints here.

## Impact:

I believe that this work, along with the proposed future work outlined in Section 6, could be impactful, provided that the ideas and methodologies are clearly presented.

## Related Work:

I did not notice any significant omissions. However, there are some NLP papers that utilize alternative methods to reduce the full training dataset, such as [this one](https://arxiv.org/abs/2009.10795). It might be worth exploring—not necessarily as related work, but as an interesting point of reference.

**Subjective Part Of Review:**

The problem is relevant to ICTIR, and the methods are somewhat original. The results are mostly negative, which is fine, but it would be nice if the authors could provide more insights. The paper itself is not easy to read and understand based on the current form.

---

### Official Review · Reviewer_mii2 · 2024-05-17

**Rating:** 2
**Confidence:** 4

**Objective Part Of Review:**

This paper investigates an important problem of distillation vs sampling in LTR. As LTR data are more complicated than common machine learning data (e.g. image classification), the authors propose a reasonable solution for distilled LTR data construction. The authors conduct thorough experiments on the proposed method.

Question:

In line 369, the authors state that "Conversely, since the majority of documents per query are irrelevant, most of the discarded documents will be irrelevant ones and highly relevant documents are more likely to be preserved". However, the statement seems incorrect, since documents are randomly sampled, regardless of their relevance. Thus irrelevant documents and relevant documents have the same chances to be preserved/discarded.

**Subjective Part Of Review:**

The presentation is good and the paper is easy to read. The proposed methods are original and promising.

---

### Official Review · Reviewer_byoZ · 2024-05-19

**Rating:** 1
**Confidence:** 4

**Objective Part Of Review:**

Overall the paper is well-written: It clearly states the problem (is dataset distillation more useful than dataset sampling?), describes the methods (the authors adapt well-known dataset distillation techniques to the learning to rank task) and analyzes experimental results.

In their experiments, the authors conclude that, in contrast to other learning tasks, they do not see an improvement of dataset distillation over random sampling, but a rather a deterioration. However, the paper would have benefited from some an experiment indicating a possible reason for this behavior. Simply stating "we believe this is due to ..." is honest (good), but immediately brings up the question of how this belief can be supported by any (initial) experiment.

There seem to be only very few mistakes in the paper:
* page 4, left column, line 277: Shouldn't distillation be applied to X_f or the randomly initialized dataset X_d? The attribute "synthesized" is confusing.
* page 6, right column, line 637 - 643: Don't you have 3 initialization (Q, D, L) instead of 2, but only 2 distillation algorithms (GM and DM)?

**Subjective Part Of Review:**

The paper is interesting to read and easy to understand. Dataset distillation is an important topic.
However, the results are disappointing. The proposed and investigated dataset distillation methods perform worse than random dataset sampling (i.e., worse than the proposed initialization methods for the dataset distillation). This requires further research. In principle, if the dataset distillation approaches make sense for the task at hand, they should at least achieve the same performance as the initialization. Follow-up work needs to find out what was missed. From this the perspective, the failure of successful dataset distillation is a valuable result for the learning to rank community.

* Figure 2-4: The performance drop of random sampling initialization ("RV") with the largest relative dataset size for Istella-S seems to be strange. -How can this happen systematically

---

### Meta-Review · Area_Chair_4Xzq · 2024-05-30

**Recommendation:** Accept (Oral)
**Confidence:** 4

**Metareview:**

In the paper, the authors ask whether dataset distillation techniques from other fields can help in learning to rank scenarios. In the empirical results, no improvement over random sampling is observed.

Negative results can be really interesting. Still, several of the reviewers have some doubts with respect to the generalizability of the observations ("concern regarding their adaptability to other LTR models"; "if the authors could provide more insights"; "Follow-up work needs to find out what was missed"). Thus, the paper under review is not the strongest possible contribution on the topic (e.g., experimenting just with a three-layer MLP). Further analyses on the generalizability for other models would definitely have strengthened the paper but I think that the current status already can be an interesting starting point for further investigation and also for discussions at the conference. As such discussions will probably be more natural at a poster booth, I suggest acceptance as a poster.